# Spatiotemporal Distribution of Dengue and Chikungunya in the Hindu Kush Himalayan Region: A Systematic Review

**DOI:** 10.3390/ijerph17186656

**Published:** 2020-09-12

**Authors:** Parbati Phuyal, Isabelle Marie Kramer, Doris Klingelhöfer, Ulrich Kuch, Axel Madeburg, David A. Groneberg, Edwin Wouters, Meghnath Dhimal, Ruth Müller

**Affiliations:** 1Institute of Occupational Medicine, Social Medicine and Environmental Medicine, Goethe University, 60590 Frankfurt am Main, Germany; Kramer@med.uni-frankfurt.de (I.M.K.); klingelhoefer@med.uni-frankfurt.de (D.K.); kuch@med.uni-frankfurt.de (U.K.); magdeburg@med.uni-frankfurt.de (A.M.); groneberg@med.uni-frankfurt.de (D.A.G.); meghdhimal2@gmail.com (M.D.); ruth.mueller@med.uni-frankfurt.de (R.M.); 2Institute of Environment and Sustainable Development, University of Antwerp, 2000 Antwerp, Belgium; 3Department of Sociology, University of Antwerp, 2000 Antwerp, Belgium; edwin.wouters@uantwerpen.be; 4Health Research Section, Nepal Health Research Council, Ramshah Path, Kathmandu 44600, Nepal; 5Unit Entomology, Institute of Tropical Medicine, 2000 Antwerp, Belgium

**Keywords:** epidemics, monsoon, postmonsoon, public health

## Abstract

The risk of increasing dengue (DEN) and chikungunya (CHIK) epidemics impacts 240 million people, health systems, and the economy in the Hindu Kush Himalayan (HKH) region. The aim of this systematic review is to monitor trends in the distribution and spread of DEN/CHIK over time and geographically for future reliable vector and disease control in the HKH region. We conducted a systematic review of the literature on the spatiotemporal distribution of DEN/CHIK in HKH published up to 23 January 2020, following Preferred Reporting Items for Systematic Reviews and Meta-Analysis (PRISMA) guidelines. In total, we found 61 articles that focused on the spatial and temporal distribution of 72,715 DEN and 2334 CHIK cases in the HKH region from 1951 to 2020. DEN incidence occurs in seven HKH countries, i.e., India, Nepal, Bhutan, Pakistan, Bangladesh, Afghanistan, and Myanmar, and CHIK occurs in four HKH countries, i.e., India, Nepal, Bhutan, and Myanmar, out of eight HKH countries. DEN is highly seasonal and starts with the onset of the monsoon (July in India and June in Nepal) and with the onset of spring (May in Bhutan) and peaks in the postmonsoon season (September to November). This current trend of increasing numbers of both diseases in many countries of the HKH region requires coordination of response efforts to prevent and control the future expansion of those vector-borne diseases to nonendemic areas, across national borders.

## 1. Introduction

Dengue (DEN) is one of the fastest spreading infectious human diseases of the twenty-first century, and chikungunya (CHIK) is an emerging public health threat worldwide [1]. DEN is caused by the dengue virus (DENV), which is distinguished in 4 serotypes, DENV-1 to -4 and CHIK, by the chikungunya virus (CHIKV) [2,3]. According to estimates of the World Health Organization (WHO), around 100 million DEN infections occur worldwide annually, and approximately 2.5 billion of the world’s population live in DEN-endemic areas [4]. Thus, it has a major socioeconomic and public health impact on the epidemic regions [5]. As CHIKV/DENV/malaria share almost the same geographic areas and show similar clinical signs and symptoms, including fever, headache, nausea, and, in a few cases, hemorrhage, it is difficult to distinguish between these vector-borne diseases by clinical symptoms alone [4,6,7]. Due to this similarity of symptoms, misdiagnosis and under-reporting of actual DEN/CHIK cases in malaria-endemic areas are very common [7].

DENV and CHIKV are vector-borne diseases (VBDs) that are transmitted to humans by the mosquitoes *Aedes aegypti* and *Aedes albopictus* [4]. The distribution of these VBDs is generally determined by a complex dynamic of environmental and social factors [8]. Rapid unplanned urbanization, massive increases of international travel and trade, different agricultural practices, and other environmental changes can favor the new establishment and spread of vectors and can place healthy populations of nonendemic regions at risk [9]. In addition, vector control programs (e.g., vector surveillance, source reduction, elimination of container habitats) and socioeconomic (income, education, gender, education), medical (drug resistance), and climatic (seasonal weather variation, climatic variability, climate change) factors are highly likely to influence the epidemiology of VBDs [10].

The Hindu-Kush Himalayan (HKH) region is among the most diverse regions of the world in terms of environmental, sociocultural, and economic aspects [11]. This region covers a wide range of lowlands to highlands, extending from Afghanistan in the west to Myanmar in the east and includes all of Nepal and Bhutan and the mountainous areas of Afghanistan, Bangladesh, China, India, Myanmar, and Pakistan [11,12]. In 2017, approximately 240 million people were living in the HKH region, and, in 2030, the population is predicted to increase to ~300 million (www.icimod.org). Anthropogenic climate change, along with rapid landscape and demographic changes, is altering the environment of the HKH region dramatically, causing the shifting of disease vectors and disease transmission from tropical into temperate regions and highlands [12,13,14]. The literature reports outbreaks of VBDs from new areas of the HKH region and also a higher number of VBD cases, for example, in Nepal, the outbreak of 2019 [15]. These trends of increasing outbreaks and frequent reports have prompted us to conduct a study on the spatiotemporal patterns of DEN/CHIK in the HKH region. Hence, the objective of this systematic review is to monitor trends in the distribution and spread of DEN/CHIK over time and geographically in the HKH region.

## 2. Methods

A systematic literature review was performed in order to summarize information on the spatiotemporal distribution of DEN/CHIK in the HKH region, following the Preferred Reporting Items for Systematic Reviews and Meta-Analysis (PRISMA) guidelines. All Web of Sciences (WoS) databases (Web of Science Core Collection, Biological Abstracts, BIOSIS Citation Index, Current Contents Connect, Data Citation Index, Derwent Innovations Index, KCI-Korean Journal Database, Medline, Russian Science Citation Index, SciELO Citation Index, and Zoological Record) were searched for peer-reviewed articles in the English language published up to 23 January 2020. The following search terms were applied in title, abstract, and keywords (topic search) in order to generate the database with all relevant articles related to our topic listed in WoS (full search term available in Appendix A)
VBDs (DEN, CHIK) and their synonyms and related insect vectors [AND].Names of the countries in the HKH region or names of territories of countries, as well as river and mountain areas in the HKH region (given by the International Centre for Integrated Mountain Development (ICIMOD) [16]), and their synonyms were added.

Inclusion criteria for selecting studies to our final database were
Epidemiological studies dealing with both spatial and temporal distribution of DENV or CHIK.Studies conducted in the HKH region countries (as defined by the International Centre for Integrated Mountain Development (ICIMOD) [16]).Studies published up to 23 January 2020.

All articles not matching the inclusion criteria and other non-peer-reviewed articles were excluded (Appendix A). For final eligibility, all selected and rejected articles were verified by a second person. Subsequently, the final database was reported (Figure 1). All included articles were downloaded and analyzed based on the objective of this study.

### Spatiotemporal Distribution of DEN and CHIK

The final database (Appendix A)was analyzed by the publication year and the countries of origin to build up time- and country-specific bibliometric figures. For the systematic review, the database was sorted by DEN, CHIK, and their vectors (Appendix A):DEN and *Aedes*CHIK and *Aedes*DEN/CHIK and *Aedes*

In total, 66 articles were qualitatively analyzed by the year of publication, the study location, the study period, the disease reported month, and the peak month for disease outbreak (Table 1 and Table 2). Among those 66 articles, 61 articles giving the exact number of DEN/CHIK cases (Appendix A) were quantitatively analyzed for the number of reported and confirmed DENV/CHIK cases and, finally, the spatiotemporal distribution of DEN and CHIK observations in HKH from 1951 to 2020.

## 3. Results

### 3.1. Bibliometric Description of Database

As shown in the PRISMA flow diagram (Figure 1), the screening and eligibility check of 490 initially searched records (Appendix A) resulted in a database with 66 original articles (Appendix A), that fully met the inclusion criteria. Out of these 66 selected articles, 83.33% dealt with the investigation of DEN (*n* = 55), 10.61% with CHIK (*n* = 7), and 6.06% with the investigation of both DEN and CHIK diseases (*n* = 4). As shown in Table 1, the articles mainly address epidemiological and entomological studies, and the applied methods are mainly descriptive or model-based, out of which six articles are case reports. Besides epidemiological studies on DEN and CHIK, entomological surveys have been carried out in 19 studies. Therein, the responsible vectors for the transmission of DEN and CHIK were *Ae. aegypti* and *Ae. albopictus*. Other vectors responsible for the transmission of DEN and CHIK were not reported. Out of the 66 articles, the first article on DEN was published in 1952 and on CHIK in 1975. From 1976 to 2008, very few articles on DEN were published (*n* = 10). A steady increase of publications from 2009 onwards, on DEN and CHIK, resulted in 52 articles from 2009 to 2020. The maximum number of publications was reached in 2018 (*n* = 9) (Appendix A).

With regard to international collaboration, authors from Nepal collaborated the most with other countries and wrote sixteen articles as first authors, with coauthors from Japan (*n* = 7), China (*n* = 3), India (*n* = 2), Bangladesh (*n* = 1), Germany (*n* = 2), USA (*n* = 1), Thailand (*n* = 2), the UK (*n* = 1), and Korea (*n* = 1) (Appendix A).

### 3.2. Number of Reported DEN and CHIK Cases in the HKH Region

In total, 72,715 DEN cases were reported from all HKH countries from 1995 to 2019, from which 52.50% cases were clinically confirmed (Appendix A). For CHIK, 2334 suspected cases were reported throughout the HKH region, of which 14.40% were clinically confirmed (Appendix A). The clinical confirmation of both DEN and CHIK was done by serological tests using enzyme-linked immune-sorbent assays (ELISAs) to capture for nonstructural protein 1 antigen (NS1), immunoglobulin M (IgM), and immunoglobulin G (IgG) antibodies.

The highest number of clinically confirmed DEN cases within the HKH region was reported from Nepal (56.83%) (Appendix A),followed by India (41.91%) (Appendix A), whereas a low number of DEN cases within the HKH region was reported from Bhutan (0.58%) (Appendix A), followed by the HKH countries of Pakistan (0.36%) (Appendix A), Bangladesh (0.26%) (Appendix A), and Afghanistan (0.06%) (Appendix A). Two articles dealt with the coinfection of DEN and other diseases: one article reported a coinfection of DEN with scrub typhus [33], and another article reported the coinfection of DEN, malaria, and scrub typhus [36].

Similar to DEN, the highest number of clinically confirmed CHIK cases within the HKH region was reported from India (77.68%) (Appendix A), followed by Nepal (11.61%) (Appendix A), and Bhutan (10.71%) (Appendix A). No CHIK cases were reported from the HKH region in Afghanistan, Bangladesh, Pakistan, and China. A study of Thaung et al. [73] reported the prevalence of DEN/CHIK in Myanmar, but this study could not be included in the quantitative analysis due to a lack of information on the number of DEN/CHIK reported or confirmed cases.

### 3.3. Spatiotemporal Distribution of DEN in HKH Countries

Figure 2A shows the spatial distribution of DEN in the HKH region. DEN was first reported in India in Northern Assam in 1951 [74], in Jammu state in 1974 [72], and in Nagaland in 1994 [71]. No reports were available for the following ten years, but a number of DEN outbreaks from several HKH provinces in India were reported from 2005 to 2017: 2005 in Darjeeling [62]; 2007 and 2008 in Manipur [54,57] and in Assam [60]; 2010 in Uttarakhand and Assam [42,56]; 2011 in Jammu Province [40] and Assam [42]; 2012 in Uttarkhand [36], Jammu Province [40], Assam [42], and Arunachal Pradesh [51]; 2013 from Uttarkhand [18,36,39,50], Jammu province [40], and Assam [42]; 2014 from Uttarakhand [18,39], Assam [20], Arunachal Pradesh [26], Himanchal Pradesh [28], and Jammu province [40]; 2015 in Assam [20], Arunachal Pradesh [21,26], Himanchal Pradesh [28], and Jammu Province; 2016 from Jorhat, Assam [24], Manipur [25], and Jammu Province [27]; 2017 from Manipur [25].

In Nepal, DEN was first reported in a Japanese volunteer working in Nepal in the year 2004 [69]. Local transmission of DEN was confirmed in the lowland areas of 11 districts (Bardiya, Banke, Dang, Salyan, Sindhuli, Birjung, Parsa, Rupandehi, Jhapa, Kapilbastu, Dhading) with a circulation of all four serotypes during the first DEN outbreak in August to November 2006 [75,76]. Afterward, DEN cases were reported from the western lowland region of Nepal in 2007 [61]. In 2008 and 2009, DEN was expanded geographically to western, far-western, and central Nepal [59]. DEN outbreaks were continuously reported from 2006 onwards and spread to 32 districts located in the lowlands, hills, and highland regions of Nepal out of Nepal’s total 75 districts [19,23]. A large DEN outbreak was reported in 2019 from 68 districts of Nepal, with more than 10,000 reported cases and six deaths [15]. Similar to Nepal, DEN in Bhutan was reported not earlier than 2004, and then again in 2005, 2006 [63], 2013, and 2014 [45].

The studies focusing on Pakistan revealed an earlier DEN infection within the HKH region in 1995 [70]. After a long gap, cases were reported again in 2007, 2009, 2010, 2011, 2012, and 2013 [48] from this HKH region. In Bangladesh, DEN was only reported in the years 2000 and 2001 [68] and in Afghanistan in the years 2010 and 2011 [35]. In the HKH region of Myanmar, DEN cases were reported quite early in 1973 and 1974 [73], but no DEN cases were reported from there after 1980 [52]. 

Figure 3 andTable 2 show the temporal distribution of DEN/CHIK in HKH regionper year and month (Appendix A), respectively. The DEN studies from India, Nepal, and Bhutan specified the monthly distribution of DEN incidence or the months when the DEN outbreak starts to occur and the peak month for the DEN outbreak [18,19,20,21,23,24,25,27,28,29,32,34,39,40,41,42,43,45,51,56,58,61,62,63,64,71,72] DEN cases in India start to occur in July [18,24,40,51,56]. However, DEN cases were reported from the month of June to December from Nepal [47,49,50,59,60,62,63,65]. In Bhutan, DEN was reported to start in May [45]. Most DEN cases in India and Nepal were found in September, October, or November [28]. However, one article from India [57] showed that the DEN outbreak peaks in December. Other literature from Nepal reports August as DEN peak month [34]. From Bhutan, July and September were reported as peak months for DEN outbreaks [45]. Deducted from this sparse information, it appears that the DEN outbreak period and the DEN peak month for DEN outbreaks vary only slightly within the HKH region. The articles related to the HKH countries of Pakistan, Afghanistan, Myanmar, and Bangladesh do not mention DEN outbreak periods and peak month for the disease outbreak.

### 3.4. Spatiotemporal Distribution of CHIK in HKH Countries

Figure 2B shows the spatial distribution of CHIK in HKH countries. CHIK cases in the HKH region were reported earliest from Myanmar in the years 1973 and 1974 [73], although no other CHIK cases were reported from Myanmar until 23 January 2020. Approximately four decades after the first CHIK report in the HKH region, additional CHIK cases were reported from India in 2008 [60], 2010 [46], and 2014 [17,26] (Figure 3C). From 2014 to 2017, CHIK cases were annually reported from India [17,26,27,31]. In Nepal, CHIK was first reported from Dhading and Kathmandu districts in 2013 [44,49], 2014, and 2015 [30] (Figure 3D). In Bhutan, CHIK cases were exclusively reported to occur in 2012 [52].

A study conducted in the HKH region of India from 2014 to 2017 [17] shows that CHIK infection occurs throughout the whole year. Other studies from the HKH region of India showed that CHIK disease peaks in September, October, and November [17,27,46]. In Nepal, CHIK cases were reported from August to November [44] and in March, May, and June [49]. However, no study from Nepal mentioned a peak month for CHIK infection. It has to be noted that CHIK cases are reported from the hilly and lowland regions of Nepal [30,44,49]. In Bhutan, CHIK cases were reported to occur in July [52], whereas, in Myanmar, no peak season of CHIK occurrence was documented [73].

## 4. Discussion

The present study determines the spatial and temporal distribution of 72,715 DEN and 2334 CHIK cases in the HKH region, as reported in the scientific literature from 1952 to 2020. The studies included in our database reveal that DEN occurs in HKH areas of at least seven out of eight HKH countries and CHIK in at least four out of eight HKH countries. DEN fever emerged in the HKH region in 1951, thus, twenty years earlier than CHIK fever. An increase of reported DEN and CHIK cases in the HKH region was observed from 2004 onwards. 

In the HKH region, DENV was first recognized in tea gardens of Northern Assam, India, in 1951 [74]. According to the literature, a large number of states (eight states and two provinces) in the Himalayan and sub-Himalayan region of India have been affected by DEN after 2005 onwards [18,20,21,24,25,27,28,33,36,39,40,42,50,51,54,56,57,60,62,65,71,72,74] The virus circulating in India in the 1950s, causing mild diseases, was replaced or evolved into genotypes with bigger virulence and transmissibility [82]. The movement of the human population from DEN-endemic areas of India to nonendemic areas might have significantly contributed to the outbreak of DEN in new areas and more frequent DEN epidemics after 2005 [57].

The DEN cases in Pakistan were earlier reported among employees of a construction contractor at the power generation plant in Baluchistan in 1995 [70], but no further DEN cases were reported in this HKH country until 2007. Lack of proper surveillance systems or complexity in laboratory diagnosis due to infection with other diseases like malaria, typhoid, and hepatitis might cause an underreporting of cases [83]. In the meantime, frequently reported DEN incidence in this region from 2007 onwards [48] might be driven, in part, by increasing human mobility, particularly in areas with climatic suitability for the mosquito vector [84]. A mobile-phone-based study conducted in Pakistan shows the mobility of infected travelers from endemic regions to all other parts of the country during the outbreak in 2013 [85]. In Nepal as well, the improvement in social and infrastructure development such as the expansion of roads, rapid urbanization, and increase in trade and business opportunities after the end of decade-long armed conflict (1996–2006) has increased the mobility of people within and from neighboring countries [86]. This fact might be the cause for the rapid expansion of DENV in this HKH country, supported by entomological investigations that show the presence of *Ae. albopictus* in Nepal in 1956 already [87]. However, the primary vector for DEN *Ae. aegypti* was reported for the first time in 2006 from the lowlands of Nepal [76], later in 2009 from Kathmandu, which is a hilly region of Nepal [88], and reports of *Ae. aegypti* and *Ae. albopictus* from at least 2000 m above sea level [89]. In parallel, cases of DEN were reported from Terai (lowland) and hill and mountain regions (68 out of 77 districts of Nepal) in 2019 [15]. Although the recent study in Afghanistan [90] reported the first locally acquired DENV cases in 2019, our study shows that dengue antibodies were detected earlier, in between 2010 and 2011, among Afghan National Army recruits in Afghanistan. However, DEN was distributed more frequently or rapidly in the HKH countries of India, Nepal, Pakistan, and Bhutan. China is the only HKH country with no reported cases of DEN in the HKH region.

The distribution of CHIK was observed in Myanmar [73], India [17,26,27,31,46,60], Nepal [30,44,49], and in Bhutan [52]. Although CHIK was observed quite early in Myanmar in 1973 and 1974, it was most frequently reported in India. However, no clear conclusions can be drawn for CHIK due to the low number of articles in our database and due to the unavailability of local governmental reports.

The period of outbreaks of DEN and CHIK varies only slightly within the HKH region. Most of studies from India [18,24,40,51,56] and Nepal [19,23,29,32,41,43,61,64] show almost consistent seasonality of DEN outbreaks, lasting from June to December. The majority of the studies from India and Nepal reported maximum DEN cases in September, October, and November in the postmonsoon period [18,19,20,23,24,25,27,28,29,39,40,41,42,43,54,61,62]. Our findings concerning a seasonal distribution of DEN in HKH countries are similar to the studies on Brazil [91] and Thailand [92]. These findings are further supported by entomological studies conducted in Nepal [89] and in Assam, India [93], that show a high abundance of vectors *Ae.*
*aegypti* and *Ae.*
*albopictus* at the end of the monsoon and postmonsoon season (September to November) compared to the winter season. The peak for DEN transmission during the postmonsoon season might be due to the most favorable weather conditions, including moderate rainfall, mild mean temperature, and optimum temperature range, which help vectors to breed, survive, and reproduce [89]. Temperature fluctuations also influence DEN infection in mosquitos [94]. Accordingly, adult *Ae. aegypti* live longer and, thus, were more likely to become infected under moderate temperature fluctuations. Moderate temperature fluctuations are typical during high DEN transmission season, whereas large temperature fluctuations favor a low dengue transmission season [94]. In *Ae. albopictus*, a temperature fluctuation from 28°-23°-18°C showed a probability of lower DEN transmission (in regard to virus titer in the salivery glands) than a constant 28 °C temperature [95]. Accordingly, vector activity and, thus, virus distribution are linked since temperatures influence, especially during the DEN season, virus transmission of the mosquitos.

The scientific reports of DEN and CHIK outbreaks in the HKH region have gradually increased from 2004/2005 until 2020 (Appendix A). The reported incidence rates show an increase in disease burden over time (Figure 3). Accordingly, this increasing trend indicates the expansion of these disease vectors in the HKH region. The distributional shifts and increasing abundance of those disease vectors are also directly or indirectly influenced by weather variables such as temperature, humidity, and precipitation [57]. The Intergovernmental Panel on Climate Change (IPCC) also concludes that anthropogenic climate change, in particular, the changing temperature and precipitation patterns, has already altered the distribution of vector-borne diseases worldwide [96]. Nevertheless, the impacts of climate change on the distribution of DEN vectors should always be considered in the context of multiple social, behavioral, economic, environmental, and health system factors. For instance, human mobility from DEN endemic areas to non-DEN-endemic areas is vital for determining the changes in exposure and susceptibility to the DEN virus in the face of climate variability and change [97]. Furthermore, problems with water scarcity and, consequently, human behavior can even increase the breeding opportunities for DEN vectors [86]. The dry climate could force local communities to store water in containers, which ultimately increases the breeding sites for *Ae. aegypti* and *Ae. albopictus* [89,98].

### Limitations of the Study

We have calculated the total number of reported and confirmed DEN and CHIK cases from each HKH country, but we have certainly failed to report the true number of DEN and CHIK cases from each HKH country. More than one study was conducted in same period, especially in India [17,18,20,21,24,25,26,27,28,36,39,40,42,50,51,54,56,57,60] and in Nepal [19,22,23,30,34,37,38,41,43,44,47,49,53,55,58,59,61,64,66,67,69]. This might have duplicated the DEN or CHIK cases in our database. In addition, we extracted the data only from English publications, which could account for unexpected deficits in scientific articles from China, Afghanistan, Bangladesh, and Myanmar in our database. Correspondingly, the underreporting of DEN or CHIK cases from these countries may be based on a lack of research in those particular VBDs or a poor DEN/CHIK surveillance system in the HKH areas of these countries. Additionally, the lack of an appropriate surveillance system for the differentiation of DEN/CHIK, a lack of appropriate diagnosis tools leading to a misdiagnosis of DEN/CHIK with malaria and other vector-borne diseases, and unfamiliarity of health workers with the epidemiology of DEN/CHIK may negatively bias the number of reported cases. 

## 5. Conclusions

DEN and CHIK viruses’ expansion is widespread in the HKH region and reveals an increasing trend of infection. DEN and CHIK are highly seasonal. DEN starts with the onset of the monsoon (July in India, June in Nepal) and with the onset of spring (May in Bhutan), and CHIK occurs throughout the whole year. Similarly, both DEN and CHIK peak in the postmonsoon season, i.e., September, October, and November. Temperature and climatic changes have a high impact on the distribution of mosquitoes and viruses. Therefore, an advanced understanding of the spatiotemporal distribution of DEN and CHIK in connection with the rapid changes in climate, infrastructure, social mobility, human behavior, and vector distribution in the HKH region is important for the improvement of DEN and CHIK disease control and vector control management. For this, outbreak preparedness and response, more climate change data, modeling, and supranational studies are needed. Further, DEN and CHIK diagnostics and reporting processes need improvement. The increasing prevalence of both diseases in more provinces of the HKH region requires the need for global coordination of response efforts across national borders.

## Figures and Tables

**Figure 1 ijerph-17-06656-f001:**
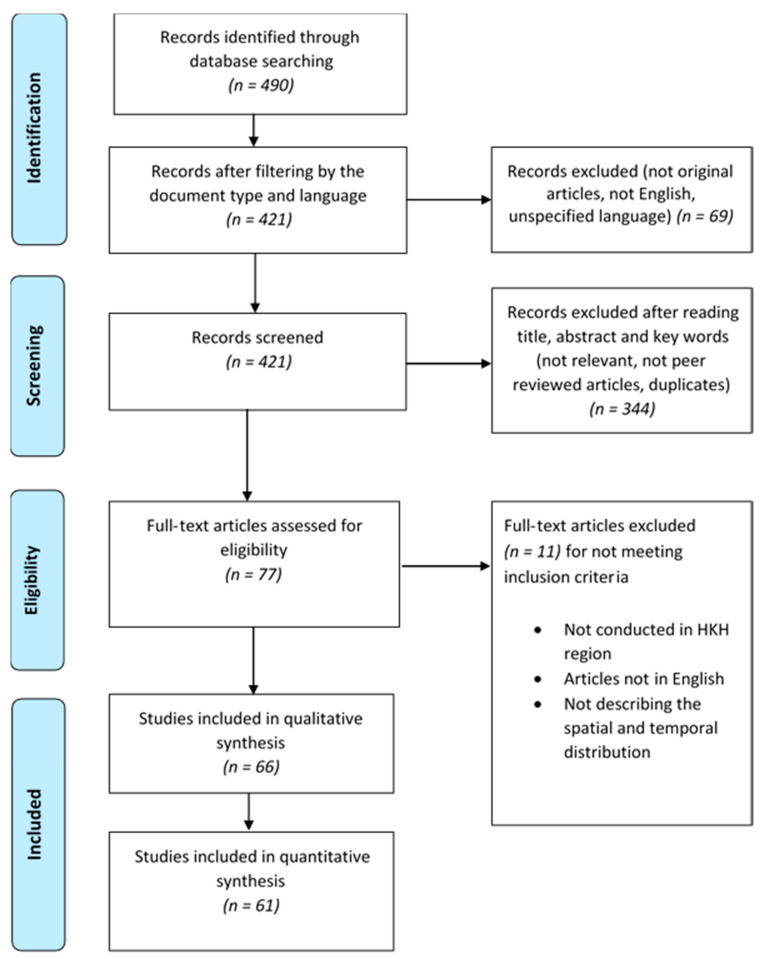
PRISMA flow diagram for generation of the database.

**Figure 2 ijerph-17-06656-f002:**
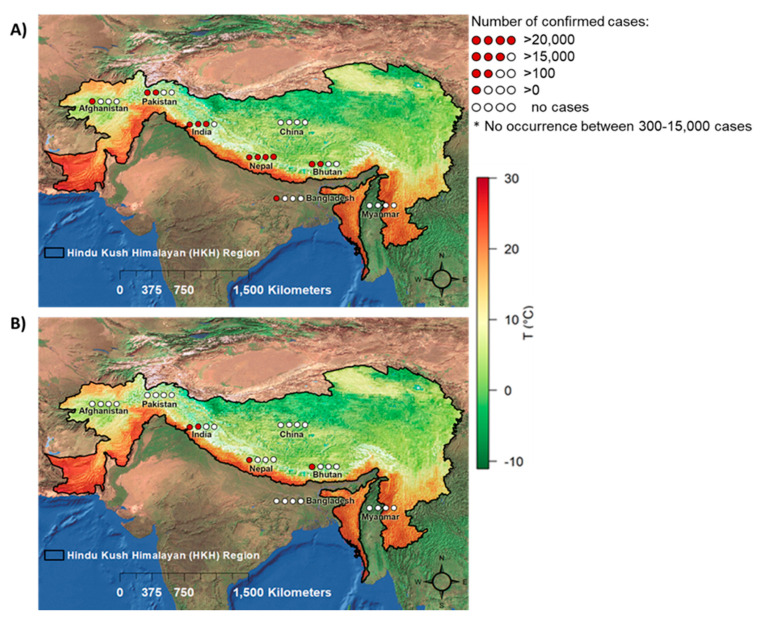
Spatial distribution map of DEN (**A**) and CHIK (**B**) in HKH. Data source: 61 publications. The average annual temperature of the HKH region is also given. The climate in the HKH region ranges from subtropical to temperate in the lower elevation areas and subalpine to alpine in the higher elevation areas.

**Figure 3 ijerph-17-06656-f003:**
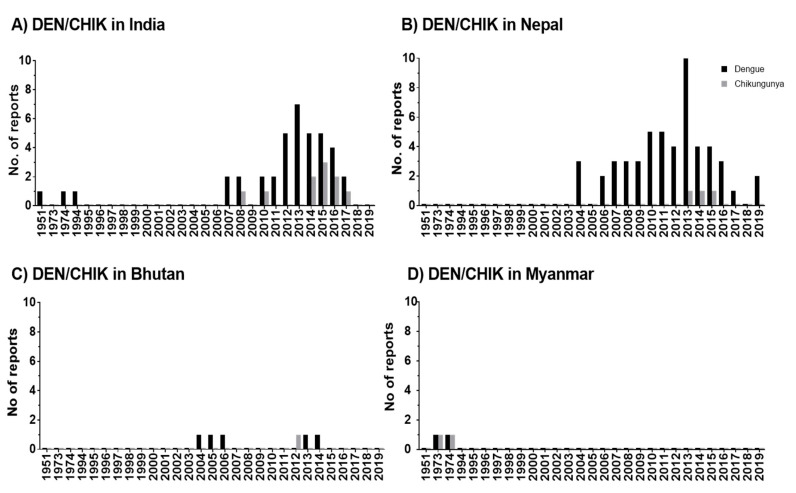
Temporal distribution of DEN and CHIK reports in the HKH region of India (**A**), Nepal (**B**), Bhutan (**C**), and Myanmar (**D**). In publications, the study has been conducted within one or multiple years. Therefore, here, the numbers are given by considering the individual study year as individual reports. Zero reports of DEN/CHIK are indicated as 0.1 value.

**Table 1 ijerph-17-06656-t001:** Overview of studies on the spatiotemporal distribution of dengue (DEN) and chikungunya (CHIK) in the Hindu-Kush Himalayan region (HKH).

Reference No.	Location (Study Period)	Diseases/Vector	Method	Main Findings
[17]	India (2014–2017)	Chikungunya and *Aedes*	Descriptive	Chikungunya cases were reported along with *Aedes aegypti* and *Ae.albopictus* from Assam, Meghalaya, and Arunachal Pradesh.
[18]	India (2012–2014)	Dengue and *Aedes*	Descriptive	Highest dengue incidence was reported in October and November
[19]	Nepal (2013)	Dengue and *Aedes*	Descriptive/Geospatial technique	Higher risk of dengue incidence was reported in postmonsoon season
[20]	India (2014–2015)	Dengue	Descriptive	Higher frequency of dengue infection was reported from September to October with circulation of DENV-1 and DENV-2
[21]	India (2015)	Dengue and *Aedes*	Descriptive/Molecular characterization	DENV-1 was reported as predominant serotype for dengue outbreak
[22]	Nepal (2011–2016)	Dengue and Aedes	Modeling	Dengue fever in Jhapa district was heterogeneously distributed and highly clustered at ward level
[23]	Nepal (2016)	Dengue	Descriptive	Dengue cases were reported in 32 districts out of Nepal’s total 75 districts, including Terai lowlands and hilly regions
[24]	India (2016)	Dengue	Cross-sectional study	Maximum number of dengue cases was reported during postmonsoon season
[25]	India (2016–2017)	Dengue	Cross-sectional study	Maximum number of dengue cases was reported in October and November (postmonsoon) in 2016 and in July and August (monsoon) in 2017
[26]	India (2014–2015)	Dengue and Chikungunya virus	Descriptive	Circulation of chikungunya virus was reported for the first time along with dengue virus
[27]	India (2016)	Dengue and Chikungunya	Cross-sectional study	Dengue and Chikungunya cases were reported in Jammu province; chikungunya was not reported prior to 2016 from this province
[28]	India (2015)	Dengue and *Aedes*	Descriptive	Dengue was confirmed as epidemic in Himalchal Pradesh with presence of *Aedes* mosquitoes
[29]	Nepal (2010)	Dengue and *Aedes*	Descriptive	DENV cases were reported from 12 districts of central Nepal and Western Nepal with circulation of DENV-1 and DENV-2 serotypes
[30]	Nepal (2014–2015)	Dengue/Chikungunya and *Aedes* mosquito	Descriptive	Dengue and chikungunya infection was reported from lowland districts of Nepal
[31]	India (2015)	Chikungunya	Descriptive	Chikungunya cases were reported from Guwahati, the capital city of Assam
[32]	Nepal (2013)	Dengue and *Aedes aegypti*	descriptive Cross-sectional	Prevalence of dengue was reported with the presence of *Aedes aegypti*
[33]	Nepal (not mentioned)	Dengue and Scrub typhus	Case report	Coinfection of dengue and scrub typhus was reported in a 50-year-old female from Chitwan district
[34]	Nepal (2010–2014)	Dengue	Descriptive	Rapid expansion of dengue fever in 32 districts out of total 75 districts was reported
[35]	Afghanistan (2010–2011)	Dengue	Descriptive	Prevalence of dengue infection among Afghan National Army Recruits was reported
[36]	India (2012–2013)	Dengue	Descriptive	Dengue reported as monoinfection and coinfection with scrub typhus and malaria
[37]	Nepal (2015)	Dengue	Case report	Report of dengue infection from a dengue nonendemic hilly district of central Nepal (elevation: 1800 m)
[38]	Nepal (2013)	Dengue	Descriptive	Dengue cases were reported from central and western Nepal
[39]	India (2013–2014)	Dengue	Descriptive	Dengue cases were recorded in monsoon and postmonsoon season
[40]	India (2011–2015)	Dengue	Descriptive	Increasing trend of dengue virus infection (2011–2015) was reported. Serum samples were collected only from governmental hospitals
[41]	Nepal (July to December 2013)	Dengue	Descriptive	Rapid expansion of dengue fever during monsoon and postmonsoon season was reported with circulation of DENV-2
[42]	India (2010–2013)	Dengue	Research note	Increasing trend (2010–2013) of dengue fever infection was reported during postmonsoon season
[43]	Nepal (2007–2013)	Dengue	Descriptive	Dengue cases reported from 12 districts of Nepal where highest cases were reported from Chitwan district (Terai lowland)
[44]	Nepal (August to November 2013)	Dengue and Chikungunya	Descriptive	Chikungunya virus reported in Nepalese patients
[45]	Bhutan (2013–2014)	Dengue	Descriptive	Dengue cases reported from two southern districts of Bhutan with circulation of DENV-1, DENV-2, DENV-3. Higher cases were reported in summer season
[46]	India (2010)	Chikungunya and Aedes	Descriptive	First reported outbreak of chikungunya from Meghalaya, Northeast India with record *of Aedes albopictus* and *Aedes aegypti* mosquitoes
[47]	Nepal (2011)	Dengue	Cross- sectional study	Report of dengue cases from two hospitals of Nepal, i.e., Rapti Zonal Hospital, Dang and Bharatpur Hospital, Chitwan
[48]	Pakistan (2007–2013)	Dengue	Descriptive	Dengue cases were reported but the study was mainly focused on Crimean–Congo hemorrhagic fever virus (CCHFV)
[49]	Nepal (2013)	Chikungunya	Case report	The first reported chikungunya virus (CHIKV) infection in Nepal
[50]	India (2013)	Dengue	Descriptive	Dengue infected cases were reported. Patients less than 12 years old were excluded in the study
[51]	India (2012)	Dengue and *Aedes albopictus*	Descriptive	Dengue outbreak was reported from Pasighat hill station with record of *Aedes albopictus*.
[52]	Bhutan (2012)	Chikungunya	Descriptive	First reported chikungunya outbreak in Bhutan
[53]	Nepal (2010)	Dengue	Descriptive	Dengue patients with critical phase were reported during 2010 dengue outbreak.
[54]	India (2007)	Dengue and *Aedes*	Outbreak investigation	First widespread dengue outbreak reported in Northeast India
[55]	Nepal (2010)	Dengue fever and *Aedes aegypt*	Descriptive	Dengue outbreak was reported from Terai lowlands to highlands of Nepal along with the presence of *Aedes aegypti*
[56]	India (July–November 2010)	Dengue	Descriptive	Liver dysfunction and secondary infection in dengue was reported as a cause of increasing morbidity
[57]	India (2007–2008)	Dengue	Descriptive	Report of dengue outbreaks from previously dengue free northeastern state of India with circulation of DENV-2
[58]	Nepal (June–September 2009)	Dengue	Descriptive	Indigenous dengue cases reported from central and western Nepal
[59]	Nepal (2008–2009)	Dengue	Descriptive	Report of geographical expansion of dengue virus to new areas (western and far-western Nepal)
[60]	India (June–September 2008)	Dengue, Chikungunya and Aedes mosquito	Descriptive	First reported cases of chikungunya virus infection in northeast India with record of both mosquito vectors *Ae. aegypti and Ae. albopictus.*
[61]	Nepal (2007–2008)	Dengue	Descriptive	Dengue cases reported from western Terai region (lowland) with the majority of cases during postmonsoon season
[62]	India (October–November 2005)	Dengue and *Aedes sp*	Outbreak investigation	Dengue outbreak reported first time from Darjeeling district with circulation of DENV-2. (*Aedes* mosquitoes were recorded from infected areas
[63]	Bhutan (2004–2006)	Dengue	Descriptive	Three dengue serotypes DENV-1, DENV-2, DENV-3 reported circulating during dengue outbreaks (2004–2006) in Bhutan
[64]	Nepal (August–December 2007)	Dengue	Cross-sectional study	Higher prevalence of dengue infection reported
[65]	India (2005)	Dengue	Case report	The first reported case of dengue encephalitis from Arunachal Pradesh
[66]	Nepal (2004)	Dengue	Case report	Imported dengue case from Nepal reported in Japan with isolation of DENV-2
[67]	Nepal (2002–2004)	Dengue and others	Descriptive	Dengue was reported in Kathmandu.
[68]	Bangladesh (2000–2001)	Dengue	Descriptive	Dengue infection reported in children. Majority of the cases were secondary dengue infection
[69]	Nepal (2004)	Dengue	Case report	The first reported case of dengue virus infection in Nepal and the case was a Japanese volunteer in Nepal
[70]	Pakistan (1995)	Dengue	Descriptive	Expansion of epidemic dengue viral infections reported in Pakistan with the circulation of multiple serotypes of dengue
[71]	India (1994)	Dengue and *Aedes*	descriptive	First report of hemorrhagic manifestation associated with DEN-4 serotype recorded from northeastern region of India with record of *Ae. aegypti* and *Ae. albopictus*
[72]	India (August to September 1974)	Dengue	Descriptive	Major involvement of dengue virus reported during febrile epidemic in Jammu
[73]	Burma (1973–1974)	Dengue, Chikungunya, and *Aedes*	descriptive	Wide distribution of dengue and chikungunya throughout Burma was reported
[74]	India (1951)	Dengue and *Aedes albopictus*	Descriptive	Dengue was reported from northern Assam Tea gardens with record of secondary vector of dengue, *Aedes albopictus*
[75]	Nepal (2006)	Dengue	Descriptive	Cases of dengue fever and dengue hemorrhagic fever were reported from Terai (lowland) regions of Nepal
[76]	Nepal (2006)	Dengue	Descriptive	All 4 dengue serotypes were confirmed during first dengue outbreak in Nepal
[77]	Nepal (2019)	Dengue	Descriptive	Israeli travelers were diagnosed with dengue in Kathmandu with circulation of DENV-2 and DENV-3
[15]	Nepal (2019)	Dengue	Descriptive	Huge dengue outbreak was reported from 68 out of 77 districts in Nepal, with more than 10,000 cases
[78]	India (2016)	Dengue	Cross-sectional study	High prevalence of IgG antibodies to dengue was reported in blood donors during outbreak
[79]	India (2017)	Dengue	Cross-sectional study	Wide geographical variation in the distribution of primary and secondary dengue cases was reported
[80]	Nepal (2010–2017)	Dengue	Description	Dengue incidence was reported to be affected by minimum temperature at lag of 2 months, maximum temperature, and relative humidity without lag period
[81]	India (2013)	Dengue	Description	First reported outbreak of dengue fever from eastern district of Sikkim, a hilly state of northeastern India with the record of *Aedes albopictus*

**Table 2 ijerph-17-06656-t002:** Reported period of outbreak/cases and reported peak month of outbreak per country. Included countries for analysis are India, Nepal, and Bhutan. Pakistan, Afghanistan, Myanmar, and Bangladesh were excluded because data for analysis was not present for these countries.

**Country**	**Reported Period of DEN Outbreak**	**Reported Peak Month of DEN Outbreak**
**J**	**F**	**M**	**A**	**M**	**J**	**J**	**A**	**S**	**O**	**N**	**D**	**J**	**F**	**M**	**A**	**M**	**J**	**J**	**A**	**S**	**O**	**N**	**D**
India							1	1	1	1	1	1												
Nepal						1	1	1	1	1	1	1								1	1	1	1	
Bhutan					1	1	1	1	1	1	1	1									1			
	**Reported Period of CHIK Outbreak**	**Reported Peak Month of CHIK Outbreak**
**J**	**F**	**M**	**A**	**M**	**J**	**J**	**A**	**S**	**O**	**N**	**D**	**J**	**F**	**M**	**A**	**M**	**J**	**J**	**A**	**S**	**O**	**N**	**D**
India	1	1	1	1	1	1	1	1	1	1	1	1									1	1	1	
Nepal			1		1	1																		
Bhutan							1

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
