# Peer review of "Spatiotemporal Distribution of Dengue and Chikungunya in the Hindu Kush Himalayan Region: A Systematic Review"

_ijerph, 2020, doi:10.3390/ijerph17186656_

Round 1

Reviewer 1 Report

Minor edits

Comment 1: PRIZMA chart

The arrow from omitted articles from Screening to Eligibility should be removed

Comment 2

The arrow from excluded 11 articles to “Studies included in qualitative synthesis” should be removed

Comment 3

Line 147-186

This paragraph provides significant information about the geographical distribution (i.e., Province and districts) of DEN and CHIK in various countries however, Figure 2 A & 2B only shows the HKH region without specific province data indicated in the data section. It would be worthwhile for the author to recreate the make for the geographical distribution of DEN and CHIK and include country borders and highlight which province or district has had reported cases of either DEN or CHIK or both.  

Comment 4

Figure 3A-D

It would be helpful if the authors combined all four graphs into two graphs with different colors for DEN and CHIK for India and Nepal. If data is available for other countries (e.g. Bhutan), it should be also included in a new figure.  

Author Response

Comment 1: PRIZMA chart

The arrow from omitted articles from Screening to Eligibility should be removed

Response to number#1: Done as suggested

Comment 2

The arrow from excluded 11 articles to “Studies included in qualitative synthesis” should be removed

Response to Number# 2: Done as suggested

Comment 3

Line 147-186

This paragraph provides significant information about the geographical distribution (i.e., Province and districts) of DEN and CHIK in various countries however, Figure 2 A & 2B only shows the HKH region without specific province data indicated in the data section. It would be worthwhile for the author to recreate the make for the geographical distribution of DEN and CHIK and include country borders and highlight which province or district has had reported cases of either DEN or CHIK or both.  

Response to number#3: We appreciate this comment and agree that it is important to provide information on the geographical information of DEN and CHIK in various countries at the province and district levels where available. Thus, we have provided this information in the text of the data section (lines 147-186). We believe that providing the information in this form will prove more useful to the majority of users than attempting to add such relatively high-resolution data to the low-resolution maps of a very large geographic region like the HKH (Fig. 2 A + B) which in addition has quite different administrative systems and units in each of the countries it is comprised of. Besides, there are multiple disputes over national borders and territories in the HKH region whose accurate demarcation on a map of this scale could easily render it useless for epidemiological purposes.

Comment 4

Figure 3A-D

It would be helpful if the authors combined all four graphs into two graphs with different colors for DEN and CHIK for India and Nepal. If data is available for other countries (e.g. Bhutan), it should be also included in a new figure.

Response to number #4: Done as suggested. New manuscript: Page 18

Reviewer 2 Report

The authors conducted an interesting systematic review of DEN/CHIK in the Hindu Kush Himalayan Region (HKH) from 1952 to 2020, however, the writing currently feels disjointed and replete with grammatical errors, and lack of syntax. They should consider sending the manuscript to a native English speaker for proofreading.

I have some general comments on the manuscript which the authors should consider revising. I have also made comments on the PDF document. A review article generally summarizes the current state of research on a given topic, that is, it synthesizes or analyzes research already conducted in primary sources. The authors state that there is not a single article which documents the spatiotemporal patterns of these diseases in 61 the different HKH regions. This cannot be a justification for the review. The opposite is however mentioned in the abstract, so, this contradicts the results. For examples, they stated in the abstract that they found 61 articles focusing on the spatial and temporal distribution of 72,715 DEN and 2334 CHIK cases in the HKH region. Consider reconciling this information throughout the manuscript. In addition, dengue shows similar symptoms of high fever, fatigue, and nausea like malaria. It would be good to briefly mention malaria as it’s also prevalent in this region.

It was also interesting to see that DEN cases were reported earlier in Pakistan in 1995, but surprisingly, no records until 2007. Were tests carried out during this long break? The authors claim that resurgence in 2007 might have been due to increasing human movement, trade or due to climate change that permitted vectors or pathogens to shift geographical ranges into more suitable environments. Could they provide any evidence for these assertions? Are there any unpublished hospital data/reports from these countries that could also be used?

Also, authors cite that the under reported cases of DEN and CHIK might have been possible absence of vectors or may be the effectiveness of vector control measures by the local authorities, access to health care, and socio-economic development. This could easily be tested or verified from WHO reports or from local government data on vector control. Again data from local health ministries could be useful here.

How did the authors account for duplication of efforts, especially as they state that more than one study was conducted in certain years in India and in Nepal? Was there any attempt to look for papers that were not published in English? If so, how many? Could that account for missing years in certain regions?

Lastly, Table 1 is very cumbersome, so consider moving to supplementary material and just do a summary of findings in text.

Author Response

The authors conducted an interesting systematic review of DEN/CHIK in the Hindu Kush Himalayan Region (HKH) from 1952 to 2020, however, the writing currently feels disjointed and replete with grammatical errors, and lack of syntax. They should consider sending the manuscript to a native English speaker for proofreading.

Thank you so much for your suggestion. The manuscript is now checked by a native English speaker.

I have some general comments on the manuscript which the authors should consider revising. I have also made comments on the PDF document. A review article generally summarizes the current state of research on a given topic, that is, it synthesizes or analyzes research already conducted in primary sources. The authors state that there is not a single article that documents the spatiotemporal patterns of these diseases in 61 the different HKH regions. This cannot be a justification for the review. The opposite is however mentioned in the abstract, so, this contradicts the results. For example, they stated in the abstract that they found 61 articles focusing on the spatial and temporal distribution of 72,715 DEN and 2334 CHIK cases in the HKH region. Consider reconciling this information throughout the manuscript. In addition, dengue shows similar symptoms of high fever, fatigue, and nausea-like malaria. It would be good to briefly mention malaria as it’s also prevalent in this region.

Response to number #1: Thank you so much for the comments. The correction did as suggested. New Manuscript line: 38-42/ line 53-54/ line 62-64

It was also interesting to see that DEN cases were reported earlier in Pakistan in 1995, but surprisingly, no records until 2007. Were tests carried out during this long break? The authors claim that resurgence in 2007 might have been due to increasing human movement, trade, or due to climate change that permitted vectors or pathogens to shift geographical ranges into more suitable environments. Could they provide any evidence for these assertions? Are there any unpublished hospital data/reports from these countries that could also be used?

Response to number #2: Thank you so much for your comments. No records until 2007 don’t mean there were no DEN cases until 2007 in Pakistan. Cases were reported from other parts of Pakistan in 2005 but no publication and hospital data/reports were found mentioning the DEN cases between this period. A mobile-based study done in Pakistan provides the evidence of human movement and climate suitability for vector as a cause besides the epidemics of DEN. The correction did as suggested. New manuscript lines: 234-240

Also, the authors cite that the under-reported cases of DEN and CHIK might have been the possible absence of vectors or may be the effectiveness of vector control measures by the local authorities, access to health care, and socio-economic development. This could easily be tested or verified from WHO reports or from local government data on vector control. Again data from local health ministries could be useful here.

Response to number #3: We are very thankful for this suggestion. We went through some available reports of WHO and tried to find the local governmental data on vector control. After going through few available reports of WHO, we realized that this statement itself could not be the reason behind under-reported cases of DEN/CHIK in the HKH region. Therefore, we deleted the lines 329-331 in an old manuscript.

How did the authors account for duplication of efforts, especially as they state that more than one study was conducted in certain years in India and in Nepal? Was there any attempt to look for papers that were not published in English? If so, how many? Could that account for missing years in certain regions?

Response to number #4: Thank you so much for this comment. We also think that not including the articles other than the English language can result in bias and those could account for missing years or missing data too. However, we were unable to extract the data from publications other than the English language. We mentioned this limitation in new our manuscript line: 297-301. We have further edited the lines in limitations. New Manuscript lines: 296-305

Lastly, Table 1 is very cumbersome, so consider moving to supplementary material and just do a summary of findings in the text.

Response to number #5: Done as suggested. New Manuscript: Supplement table S16

Reviewer 3 Report

In this review, Phuyal et al systematically summarize the spatiotemporal distribution of DEN/CHIK in the HKH region from 1952-2020 and give information for future vector and disease control. Basically, this review is fairly completed,  but some parts still need to be further improved. Comments are listed below for authors.

1. Table 1 should be more concise and impressive. Also it would be better to change "Study (Author)" to "Reference No.".

2. Figure 2, the spatial distribution was divided into two groups (presence and absence), it would be better to divide into more detailed groups by case number.

3. In abstract and discussion, the spatial and temporal distribution of DEN and CHIK cases in the HKH region is from 1952 to 2020, in methods section, it is from 1951-2020. Authors should make this point clear. 

Author Response

Comments and Suggestions

In this review, Phuyal et al systematically summarize the spatiotemporal distribution of DEN/CHIK in the HKH region from 1952-2020 and give information for future vector and disease control. Basically, this review is fairly completed, but some parts still need to be further improved. Comments are listed below for authors.

  1. Table 1 should be more concise and impressive. Also, it would be better to change "Study (Author)" to "Reference No.".

Response to number #1: Done as suggested. New manuscript: Table on page 5-14 (Track change version) but move as supplement table S16 in a new manuscript

  1. Figure 2, the spatial distribution was divided into two groups (presence and absence), it would be better to divide into more detailed groups by case number.

Response to number #2: Done as suggested. New manuscript: Figure 2 on page 6

  1. In abstract and discussion, the spatial and temporal distribution of DEN and CHIK cases in the HKH region is from 1952 to 2020, in the methods section, it is from 1951-2020. The authors should make this point clear.

Response to number #3: Thank you for your kind comment. Actually, the first publication on dengue was in 1952, and in 1951, dengue was 1st reported according to the literature in our database. We corrected this mistake! New manuscript: line 21, Page 1